# Role of the Scavenger Receptor CD36 in Accelerated Diabetic Atherosclerosis

**DOI:** 10.3390/ijms21197360

**Published:** 2020-10-05

**Authors:** Miquel Navas-Madroñal, Esmeralda Castelblanco, Mercedes Camacho, Marta Consegal, Anna Ramirez-Morros, Maria Rosa Sarrias, Paulina Perez, Nuria Alonso, María Galán, Dídac Mauricio

**Affiliations:** 1Sant Pau Biomedical Research Institute (IIB Sant Pau), Hospital de la Santa Creu i Sant Pau, 08041 Barcelona, Spain; mnavasm7@gmail.com (M.N.-M.); Mcamacho@santpau.cat (M.C.); martaconse@gmail.com (M.C.); 2Department of Endocrinology & Nutrition, Hospital de la Santa Creu i Sant Pau & Sant Pau Biomedical Research Institute (IIB Sant Pau), 08041 Barcelona, Spain; esmeraldacas@gmail.com; 3Center for Biomedical Research on Diabetes and Associated Metabolic Diseases (CIBERDEM), 08025 Barcelona, Spain; nalonso32416@yahoo.es; 4Center for Biomedical Research on Cardiovascular Disease (CIBERCV), 28029 Madrid, Spain; 5Department of Endocrinology & Nutrition, University Hospital and Health Sciences Research Institute Germans Trias i Pujol, 08916 Badalona, Spain; aramirezm@igtp.cat; 6Innate Immunity Group, Health Sciences Research Institute Germans Trias i Pujol, Center for Biomedical Research on Liver and Digestive Diseases (CIBEREHD), 28029 Madrid, Spain; mrsarrias@igtp.cat; 7Department of Angiology & Vascular Surgery, University Hospital and Health Sciences Germans Trias i Pujol, Autonomous University of Barcelona, 08916 Badalona, Spain; paulinaperezramirez@gmail.com

**Keywords:** scavenger receptor CD36, atherosclerosis, diabetes, inflammation, vascular calcification

## Abstract

Diabetes mellitus entails increased atherosclerotic burden and medial arterial calcification, but the precise mechanisms are not fully elucidated. We aimed to investigate the implication of CD36 in inflammation and calcification processes orchestrated by vascular smooth muscle cells (VSMCs) under hyperglycemic and atherogenic conditions. We examined the expression of CD36, pro-inflammatory cytokines, endoplasmic reticulum (ER) stress markers, and mineralization-regulating enzymes by RT-PCR in human VSMCs, cultured in a medium containing normal (5 mM) or high glucose (22 mM) for 72 h with or without oxidized low-density lipoprotein (oxLDL) (24 h). The uptake of 1,1′-dioctadecyl-3,3,3′,3-tetramethylindocarbocyanine perchlorate-fluorescently (DiI) labeled oxLDL was quantified by flow cytometry and fluorimetry and calcification assays were performed in VSMC cultured in osteogenic medium and stained by alizarin red. We observed induction in the expression of CD36, cytokines, calcification markers, and ER stress markers under high glucose that was exacerbated by oxLDL. These results were confirmed in carotid plaques from subjects with diabetes versus non-diabetic subjects. Accordingly, the uptake of DiI-labeled oxLDL was increased after exposure to high glucose. The silencing of CD36 reduced the induction of CD36 and the expression of calcification enzymes and mineralization of VSMC. Our results indicate that CD36 signaling is partially involved in hyperglycemia and oxLDL-induced vascular calcification in diabetes.

## 1. Introduction

Pathologic calcification of cardiovascular structures very frequently occurs in the elderly and people suffering from diabetes and atherosclerosis and is considered an important marker of cardiovascular morbidity and mortality [1]. Atherosclerosis is a chronic inflammatory disease involving endothelial dysfunction, recruitment of inflammatory cells, local production of cytokines, and lipid accumulation within the intima of the vessel wall [2]. Inflammation and dyslipidemia play crucial synergistic roles in the deterioration of the vascular wall that occurs during the progression of atherosclerosis [3]. Although most studies on atherosclerosis have focused on the role of macrophages, vascular smooth muscle cells (VSMC) also play a decisive and, probably, understudied role. VSMC-derived foam cells in human atherosclerotic lesions are well documented [4,5], with recent studies showing that as many as 50% of foam cells in human and murine lesions are derived from VSMC [6,7]. VSMC possess remarkable plasticity with the ability to reprogram their expression pattern as contractile, synthetic, osteochondrogenic, and macrophage-like phenotypes. This is particularly relevant in human arteries, which are enriched in VSMC. Migration, proliferation, and synthesis of the extracellular matrix by VSMC contribute to early development of lesions [8]. The secretion of proliferative and inflammatory cytokines by VSMC promulgate autocrine activation of VSMC and recruitment of macrophages into the lesion in a paracrine manner [9]. Scavenger receptors mediate the uptake of oxidized low-density lipoprotein (oxLDL) by macrophages, resulting in foam cell formation that represents the hallmark of early atherosclerosis [10,11]. Regulation of lipid uptake in VSMC is poorly characterized and less understood than in macrophages, despite being a crucial event in atherogenesis [12]. Vascular smooth muscle cells express scavenger receptors for oxLDL uptake [13,14] and acquire macrophage-like phenotypes upon lipid loading by expressing macrophage markers and suppressing VSMC markers [15,16]. Because they cannot egress from the plaque, uptake of excess lipid by medial and intimal VSMC leads to plaque progression, apoptosis, and eventual plaque instability [17].

Recent work suggested that there are significant differences in gene expression and modes of lipid loading between VSMC and macrophages [18,19,20]. Studies in vitro showed lipid loading by VSMC of cholesterol and oxLDL through scavenger receptor CD36 [16,17], but mechanisms by which VSMC take up lipids in vivo are mostly unknown.

Vascular calcification (VC) is a common feature in people with type 2 diabetes mellitus (T2DM) and has been shown to be an independent marker for mortality in subjects with advanced cardiovascular diseases [21,22]. Under hyperglycemic conditions, VSMC transit markedly into osteoblast-like cells and increased calcific nodules in vitro [23]. VC induced by high glucose (HG) is considered an active pathological process that involves the crosstalk between cells and the extracellular matrix of the arteries that resembles physiological bone formation [21,24]. The process has many features comparable to embryonic bone formation and involves VSMC transdifferentiation into osteoblast-like cells [24,25]. High glucose not only induces VSMC calcification but also results in triggering inflammation and endoplasmic reticulum (ER) stress, which contributes, at least in part, to the development of VSMC calcification [23,26]. 

In this study, we aimed to investigate the contribution of CD36 to VSMC induced calcification under hyperglycemic and atherogenic conditions and the mechanisms involved in this process. 

## 2. Results

### 2.1. Hyperglycemic and Atherogenic Conditions Synergistically Induced the Expression of Scavenger Receptor CD36, and Exposure to High Glucose Enhances oxLDL Uptake in VSMC

To determine whether high glucose concentration induces CD36 expression, we incubated the VSMC with 5 mM, 15 mM, 22 mM, and 30 mM of glucose for 72 h. CD36 mRNA levels were gradually increased in response to the increasing concentrations of glucose (Appendix A). For the current study, we selected 22 mM as the HG condition since this is a concentration within ranges of poorly controlled subjects with diabetes. To determine whether exposure to HG in combination with oxLDL could induce a synergistic effect on CD36 expression, primary human VSMC were incubated with medium supplemented with 5 mM or with 22 mM for 72 h and with oxLDL for the last 24 h. We observed that the increase of CD36 mRNA levels in response to HG was significantly enhanced by the addition of oxLDL (Figure 1A). We next studied the induction of inflammatory markers in VSMC under these hyperglycemic and atherogenic conditions. In addition to CD36, monocyte chemoattractant protein-1 (MCP-1), interleukin 6 (IL-6), and interleukin 1β (IL-1β) mRNA levels were also triggered with HG, and in the case of MCP-1, the co-incubation with oxLDL potentiated the induction (Figure 1B–D).

We then investigated whether the CD36 ligand scavenger protein CD5L (also referred to as AIM), which is traditionally expressed by macrophages, could play a role in the modulation of CD36 expression and its responses to oxLDL, as it does in macrophages [27]. We found that CD5L gene expression was undetectable in VSMC cells (data not shown). When we analyzed the induction of CD36 and inflammation markers expression by HG and oxLDL in VSMC co-incubated with the recombinant protein CD5L (rCD5L) or with the same concentration of albumin as a control, we observed that rCD5L drastically reduced the CD36 gene expression in VSMC incubated with HG with or without oxLDL, but it did not affect to the expression of MCP-1, IL-6, or IL-1β (Appendix A).

ER stress, as a pathological process involved in diabetes and atherosclerosis, is induced by HG in VSMC [23]. We observed the induction of ER stress markers expression (Activating transcription factor 6, ATF6; Heat shock protein 5, HSPA5; Endoplasmic Reticulum To Nucleus Signaling 1, ERN1; and C/EBP homologous protein, CHOP) in VSMC stimulated with HG that was exacerbated by the co-incubation with ox-LDL (Figure 1E–H).

To determine whether the scavenger receptor CD36 increased expression is paralleled to an increase of oxLDL uptake in human VSMC under hyperglycemic conditions, cells stimulated with HG for 72 h were cultured with fluorescently labeled oxLDL (DiI-oxLDL) for the last 24 h. Lipid uptake, quantified by flow cytometry (Figure 2A) and by fluorimetry (Figure 2B), showed that exposure to HG significantly increased oxLDL uptake by 48% (HG DiI-oxLDL vs. NG DiI-oxLDL, Figure 2A) and by 43% (HG DiI-oxLDL vs. NG DiI-oxLDL, Figure 2B), respectively.

### 2.2. Exposure to HG Triggers Calcification in VSMC

oxLDL has been proven to play an essential role in arterial calcification development, but the mechanisms by which hyperglycemia accelerates this process remain to be clarified. We studied the effect of the exposure to HG (22 mM) with or without oxLDL in the expression of pro-calcification markers, such as alkaline phosphatase (ALPL), secreted phosphoprotein 1 (SPP1, also known as osteopontin), and bone morphogenetic protein 2 (BMP2) in VSMC. After 3 days (Appendix A) and 7 days (Figure 3A–C) of exposure to HG, we observed that the expression of ALPL, SPP1, and BMP2 was significantly increased by HG, and it was further potentiated in combination with oxLDL. High glucose-induced-CD36 expression was sustained after 7 days (Figure 3D). This was accompanied with an increase in calcium depots formation in VSMC cultured in osteogenic medium and exposed to HG for 7 days in comparison with normal glucose (NG) conditions with or without oxLDL co-incubation (Figure 3E). 

### 2.3. Silencing of CD36 Decreased ox-LDL Uptake and Limited Osteoblastic Differentiation of VSMC

We tested the mediation of CD36 on oxLDL uptake in VSMC by silencing CD36 gene expression and assessed it by mRNA levels quantification and protein expression. As depicted in Figure 4A,B, siRNA targeting CD36 abolished mRNA expression, reduced protein levels, and prevented the increase of DiI-oxLDL uptake under HG conditions by flow cytometry (Figure 4C). However, it did not alter the induction of pro-inflammatory markers expression in the absence or presence of oxLDL in comparison with VSMC transfected with siRNA CT (Figure 4D–F). 

We further studied whether silencing of CD36 affected pro-calcification markers expression and found that it was significantly reduced in human VSMC exposed to HG for 7 days with or without oxLDL in comparison with cells transfected with control siRNAs (Figure 5A–C). We then determined whether CD36 was involved in HG and oxLDL-induced osteoblastic differentiation of human VSMC cultured in osteogenic medium and exposed to NG or HG for 7 days. As shown in Figure 4D–F, HG and oxLDL treatment significantly induced calcium deposit formation assessed by alizarin red staining, and this was attenuated by CD36 silencing. However, the effects of transient silencing of CD36 were mostly lost after 10 days of culturing cells in an osteogenic medium (Appendix A). 

### 2.4. Expression of CD36 and ER Stress Indicators in Carotid Plaques from Subjects with and without Diabetes

We aimed to determine whether CD36 expression was altered in carotid plaques from subjects with diabetes (*n* = 15) when compared with plaques from subjects without diabetes (*n* = 15). Demographic and clinical data from participants included in this study are shown in Table 1. We measured gene and protein expression of CD36 by real-time PCR, immunohistochemistry, and Western blot, respectively, and observed that it was markedly increased in plaque tissue from subjects with diabetes (Figure 6A–C). We aimed to measure circulating soluble CD36 levels in plasma from the subjects included in this study, but we could not detect it in most of the samples (data not shown). 

Serum oxidized sterols (oxysterols) are a major component in oxidized LDL and are thought to mediate its lipotoxic effects. Oxysterols, such as 7-ketocholesterol (7-KC), are abundant in advanced atherosclerotic lesions [28]. Therefore, we measured the levels of free 7-KC levels in the plasma of subjects and observed that circulating 7-KC levels were significantly higher in subjects with diabetes compared to subjects without diabetes (Figure 6D).

In addition, the gene expression of ER stress markers that represent the three pathways of the unfolding protein response (UPR) was analyzed in these samples. We observed that ATF6, CHOP, ERN1, HSPA5, and ATF4 mRNA levels were significantly increased in tissues from subjects with diabetes, and this was supported by an increased expression of ATF6 and CHOP protein expression when compared with tissue from subjects without diabetes (Figure 6E–J). 

## 3. Discussion

In the present study, we show that hyperglycemia increased the expression and activity of the oxLDL scavenger receptor-CD36, induced inflammatory and ER stress markers expression, and we demonstrate the contribution of CD36 to in vitro calcification in human VSMC. These effects were synergistically enhanced by oxLDL. In addition, we found that CD36 expression was exacerbated in carotid plaques from subjects with diabetes compared with carotid plaques from normoglycemic subjects. 

In agreement with our results, in T2DM, the interplay between cells and pro-atherogenic factors has been described to upregulate CD36 expression in macrophages and endothelial cells. Together with oxLDL, raised levels of glucose, insulin resistance, low HDL cholesterol, C-reactive protein, increased oxidative stress, and plasma advanced glycation end products (AGE), all result in increased expression of CD36 and other lipids scavenger receptors, thereby contributing to T2DM and related atherosclerosis [29,30,31]. Moreover, CD36 expression has been previously reported to be altered by hyperglycemia in atherosclerotic patients [32].

Understanding the mechanisms of VC in the progress of diabetes is of high clinical relevance, above all, regarding people with diabetes suffering from atherosclerosis. Inflammation and dyslipidemia play crucial synergistic roles in the deterioration that occurs during the progression of atherosclerosis [2,3]. Type 2 diabetes is characterized by elevated serum inflammatory markers and intimal VC. Accordingly, we found an exacerbation of the expression of CD36 together with the increase in the expression of pro-inflammatory markers MCP-1, IL6 and IL-1β in VSMC exposed to HG in the absence or presence of oxLDL. Our results agree with Lopez-Carmona et al., that reported an increase of CD36 mRNA expression in peripheral blood mononuclear cells (PBMC) exposed to high glucose and observed a progressive CD36 overexpression in PBMC isolated from pre-diabetic and diabetic individuals with atherosclerosis [33].

Under inflammatory conditions, tissue macrophages secrete the protein CD5L, which increases macrophage foam cell formation and CD36-mediated oxLDL uptake [27]. CD5L has been implicated in the pathogenesis of several infections, atherosclerosis [34], and is linked to insulin resistance in obesity [35]. We did not detect CD5L expression in VSMC; however, the co-incubation of VMSC with human recombinant CD5L protein prevented the induction of CD36 suggesting that CD5L may play a potential protective role by preventing oxLDL uptake through CD36 in VSMC. Additionally, human recombinant CD5L did not modify the expression of inflammatory markers. 

The silencing of CD36 gene expression blunted the uptake of Di-oxLDL by VSMC elicited by HG. This indicates that the scavenger receptor CD36 was responsible, at least in part, for the oxLDL uptake. However, CD36 is not the only mediator of oxLDL uptake. Other scavenger receptors for lipids, such as lectin-like oxidized LDL receptor-1 (LOX-1), low-density lipoprotein receptor-related protein (LRP1), and class A scavenger receptor (SR-A), internalize oxLDL and may be upregulated in VSMC by oxLDL [14,36,37,38]. On the contrary, the silencing of CD36 did not affect the upregulated expression of pro-inflammatory markers by HG and oxLDL, suggesting that the effects observed by the exposure to HG with or without the presence of oxLDL cannot be explained by the unique action of CD36-derived signaling. HG induces the release of inflammatory cytokines via activation of the transcription factors nuclear factor kappaB (NF-kappaB) and activating protein-1 (AP-1) [39,40]. Recent studies demonstrated that the induction of MCP-1 expression by HG is mediated through the receptor of advanced glycation end-products (RAGE) via activation of NF-κB and mitogen-activated protein kinase [41,42,43]. Furthermore, HG triggers oxidative stress generation, induces cellular senescence, and leads to ER stress induction and cell apoptosis, mechanisms that are closely related to the activation of pro-inflammatory pathways in vascular cells [44,45,46]. These works support our data and help us to explain why CD36 silencing does not modulate pro-inflammatory markers expression and suggest that the effects of exposure to HG on MCP-1, IL-1β, and IL6 expression may be through the activation of RAGE. 

Besides inflammation, chronic induction of ER stress plays a key role in the development of atherosclerosis in vascular tissues and type 2 diabetes in experimental models [23,47,48]. Unfolding protein response (UPR) signaling in response to ER stress contributes to vascular cell death and VC [49,50]. In particular, CHOP-mediated cell death is widely characterized in atherosclerosis and is a crucial contributor to the development of advanced atherosclerotic lesions [50,51]. Accordingly, we found that CHOP and other ER stress markers were upregulated in VSMC after chronic exposure to HG and oxLDL and in atherosclerotic plaques from subjects with diabetes compared to those without diabetes. In line with our results, other authors previously reported that the UPR is activated in cholesterol-loaded macrophages and that ER stress inhibition effectively reduced foam cell formation and apoptosis through the decrease in the expression of CD36 and oxLDL uptake [19,52].

The mechanisms underlying the molecular pathogenesis of VC in diabetes are extremely complex and are influenced by the interaction among several inflammatory factors, lipids, and oxidative stress. It is well documented that oxLDL is a critical risk factor for atherosclerosis and osteogenic differentiation of VSMCs, which may induce osteogenesis through the activation of toll-like receptor 4 (TLR4) derived signaling via NK-κB or through the activation of the transforming growth factor (TGF)-β pathway [53,54]. However, there are no studies describing the combined effects of HG and oxLDL on VSMC mineralization. The expression of ALPL, SPP1 and BMP2 was induced in VSMC after chronic exposure to HG with or without oxLDL, indicating a transdifferentiation process into osteoblast-like cells. In agreement with previous studies [22,23,24], we observed that calcification of VSMC cultured under hyperglycemic conditions was higher than in those cultured under NG conditions in the presence or absence of oxLDL after 7 or 10 days growing in an osteogenic medium. Interestingly, the silencing of CD36 did not significantly reduce only the expression of calcification markers but also limited the induced-calcification on VSMC, as shown by alizarin red staining. Taken together, these data suggest that CD36 participates in hyperglycemia and oxLDL-induced osteoblastic differentiation of VSMC. Supporting our data, Staines et al. showed that CD36 was among novel differentially expressed genes during the process of bone mineralization in primary osteoblasts, and its upregulation was coupled with a significant increase in mRNA expression of key genes associated with osteoblast mineralization [55].

Oxysterols induce not only apoptosis of vascular cells and macrophages but also induce calcification and osteoblastic differentiation of VSMC [56]. Recent publications emphasize the importance of circulating oxysterols in the development of cardiovascular diseases, and in particular, the augmentation of 7-KC levels has been associated with cardiovascular events [57]. The circulating 7-KC levels in the subjects cohort suggest that the diabetic condition is worsening the oxysterols profile in blood, and this may also contribute to accelerate the calcification process of arteries. 

In conclusion, our findings point out an important role of CD36 in mediating the deleterious effect of hyperglycemia and atherogenic lipid molecules on the inflammatory and calcification processes of VSMC involved in the accelerated atherogenesis accompanying these metabolic conditions. Most importantly, the present study suggests that CD36 and ER stress could be potential targets for novel therapeutic strategies to alleviate vascular calcification in subjects with diabetes.

## 4. Methods

### 4.1. Human Samples

In this single-center, observational study, plasma samples and carotid plaques from symptomatic (cerebrovascular disease event) and asymptomatic subjects undergoing carotid surgical endarterectomy for carotid disease performed at University Hospital Germans Trias i Pujol (Badalona, Spain) were collected between March 2014 and March 2018. This research was approved by the institutional ethics committee of Hospital Germans Trias i Pujol and conducted in accordance with the Declaration of Helsinki; all participants signed informed consent forms. Demographics, risk factors, and clinical characteristics were recorded for all subjects. In this study, we included 36 subjects who were stratified into two groups: subjects with diabetes (*n* = 18) and subjects without diabetes (*n* = 18). Only 15 samples of tissue embedded in OCT were available per group.

All carotid endarterectomies were performed to relieve significant carotid stenosis, as established by the North American Symptomatic Carotid Endarterectomy Trial (NASCET), as previously published [58,59]: 50–99% luminal narrowing in people with symptomatic carotid stenosis and ≥ 70% in those with asymptomatic carotid stenosis. 

### 4.2. Cell Culture

Human VSMC were isolated from the abdominal aorta of multi-organ donors by an explant procedure, as previously described [60]. Briefly, endothelium-denuded medial tissue was cut into 2-4 mm cubes that were transferred to a 25 cm^2^ culture flask containing 5 mL of pre-warmed culture medium M199 (Gibco, Carlsbad, CA, USA) supplemented with 10% foetal bovine serum (FBS) and antibiotics (100 U/mL penicillin G and 100 μg/mL streptomycin) (all from Biological Industries, Kibbutz Beit-Haemek, Israel). VSMC migrate out from the explants within 2–3 weeks. The medium was exchanged every 3 days after the onset of cell outgrowth. When a significant outgrowth was reached, tissue fragments were collected with forceps and placed in a new dish with fresh medium. Then, after removing the explants from the flask surface, the cells that remain in the dish were cultured until confluence, trypsinised, used as P1 stage cells, and routinely subcultured. Cells used in the present experiments were between the third and sixth passage. VSMC at these passages appeared as a relatively homogeneous cell population, showing a hill-and-valley pattern at confluence. Western blot analysis for specific differentiation markers revealed a clear positive band for α-actin (45 kDa) and calponin (33 kDa). Human VSMC were cultured in medium M199 supplemented with 20% FBS, 2% human serum, 2 mmol/L L-glutamine (Invitrogen), 100 U/mL penicillin G, and 100 μg/mL streptomycin. VSMC from 6 different donors were used. The research was performed in accordance with the Declaration of Helsinki and approved by the Ethics Committee of Hospital de la Santa Creu i Sant Pau (12/031/1316). 

For experimental procedures, cells were seeded in multiwell plates, and subconfluent cells were starved in a medium supplemented with 1% FBS for 24 h before the addition of a medium containing 5 mM (NG, normal glucose), 22 mM of glucose (high glucose, HG,) (D-Glucose G7021, Sigma–Aldrich) or vehicle (PBS) for 72 h in the presence or absence of 50 μg/mL human purified oxLDL (BT-910 Alfa Aesar J65591) for the last 24 h. Treatments did not induce cytotoxicity analyzed by the MTT assay (Roche Diagnostics, Indianapolis, IN, USA).

In additional assays, the cells were co-incubated with 1 μg/mL endotoxin-free recombinant human CD5L (R&D systems, Abingdon, UK) or with 1 μg/mL albumin as control (Sigma–Aldrich) for 48 h.

### 4.3. Transfections with Small Interfering RNA (siRNA)

We used siRNAs against CD36 and siRNAs control (SMARTpool: ON-TARGET plus CD36 siRNA, L-010206-00-0005, and ON-TARGET, plus non-targeting siRNA #1, D-001810-01-05, Dharmacon (Thermo Fisher Scientific, Waltham, MA, USA) to transfect VSMCs. Lipofectamine^TM^ RNAiMAX transfection reagent (Invitrogen) was used for siRNA delivery. Cells were transfected with 20–50 nM siRNA, using 7.5 μL of Lipofectamine RNAiMAX Reagent following the manufacturer’s instructions. After transfection (24 h), the medium was replaced, and cells were washed twice in culture medium and incubated for at least 8 h in complete fresh medium. Thereafter, VSMC were serum-deprived for 24 h and exposed to NG or HG for 72 h with or without oxLDL for the last 24 h. In calcification experiments and after transfection, VSMC were incubated with an osteogenic medium for 7–10 days.

### 4.4. Total mRNA and Protein Isolation from Tissue and Cells

RNA and protein lysates were obtained from frozen OCT-embedded tissue by previously dissolving it in ddH_2_O to remove the OCT by centrifugation. Total RNA isolation from human VSMC and total RNA and protein isolation from human carotid plaques were performed using a tissue homogenizer and the Tripure reagent (Roche Diagnostics) following the manufacturer’s instructions. RNA integrity was determined by electrophoresis in agarose gels and was quantified by a NanoDrop 1000 Spectrophotometer (Thermo Scientific, Waltham, MA, USA). Protein lysates from cells were prepared in a RIPA buffer (150 mM NaCl, 1% (*v*/*v*) Triton X-100, 0.5% (*w*/*v*) sodium deoxycholate, 0.1% (*w*/*v*) SDS, 2 mM EDTA, 50 mM Tris-HCl (pH 8), and following a standard protocol. 

### 4.5. Quantitative Real-Time PCR

DNase I-treated total RNA (1 μg) was reverse transcribed into cDNA using the High Capacity cDNA Archive Kit (Applied Biosystems, Foster City, CA, USA) with random hexamers. Quantification of mRNA levels was performed by real-time PCR using pre-designed validated assays (TaqMan Gene Expression Assays; Applied Biosystems) for human CD36 (Hs00169627_m1), ATF6 (Hs00232586_m1), CHOP (Hs99999172_m1), HSPA5, also referred to as GRP78 (Hs99999174_m1), ERN1, also referred to as IRE1 (Hs00176385_m1), ATF4 (Hs00909569_g1), MCP-1 (Hs00234140_m1), IL1β (Hs01555410_m1), IL6 (Hs00174131_m1), ALPL (Hs01029144_m1), BMP2 (Hs00154192_m1), SPP1 (Hs00959010_m1), and CD5L (Hs00935902_m1). As endogenous controls glyceraldehyde 3-phosphate dehydrogenase (GAPDH; Hs02758991_g1) and β-actin (Hs99999903_m1) were used. Each sample was amplified in duplicate. Similar results were obtained after normalization to either housekeeping gene. Quantitative RT-PCR was carried out in an ABI PRISM 7900HT Sequence Detection System (Applied Biosystems). Relative mRNA levels were determined using the 2^-∆∆Ct^ method.

### 4.6. OxLDL Uptake

To generate DiI-oxLDL, a stock solution of the fluorescent probe: 1,1′-dioctadecyl-3,3,3′,3-tetramethylindocarbocyanine perchlorate (DiI, Molecular Probes Invitrogen, Carlsbad, CA) was prepared in DMSO and then added to the LDL solution to yield a final ratio of 1 μg of DiI/mg of oxLDL. The mixture was incubated for 18 h at 37 °C under light protection, as previously described [61]. Cells were incubated with HG or NG in M199 medium containing 0.2% SBF (*v*/*v*) for 72 h and with DiI-oxLDL (5 µg/mL) for the last 24 h. To test the specificity of uptake, the VSMC were treated with DiI-oxLDL and an excess of unlabeled oxLDL. After treatments, cells were washed three times with PBS and harvested with trypsin solution (0.25% trypsin, 0.02% EDTA). Then, the cells were centrifuged at 500× *g* for 5 min, washed once with M-199 complete medium at 37 °C and twice with PBS at 500× *g* for 5 min at 4 °C. The uptake of DiI-oxLDL by VSMC was analyzed by flow cytometry on a FACScan flow cytometer (BD Biosciences) with 10,000 events acquired for each sample. Data were calculated and expressed as mean fluorescence intensity (MFI). As negative control, VSMC incubated with un-labeled oxLDL were used. 

In parallel, the uptake of DiI labeled oxLDL was quantified by fluorimetry. After VSMC incubation with DiI-labeled oxLDL or with unlabeled oxLDL, as stated above, cells were washed 4 times with PBS and solubilized with 250 μL of 1% Triton X-100 in PBS. Culture dishes were shaken for 15 min at room temperature (RT), and the supernatant was removed. Fluorescence was measured immediately at 530 nm excitation-light and 590 nm emission-light. 

Additional oxLDL uptake experiments were performed after the silencing of CD36 expression. Briefly, VSMC were transfected with 20 nM of a set of four siRNAs targeting CD36 or an equal concentration of a non-targeting negative control pool by using Lipofectamine^TM^ RNAiMAX and following the manufacturer’s instructions. Twenty-four hours later, the medium was replaced, and cells were incubated with HG or NG in M199 medium, containing 0.2% SBF (*v*/*v*) for 72 h and with DiI-oxLDL for the last 24 h. 

### 4.7. In Vitro Calcification 

Calcification of human aortic VSMC was induced by culturing confluent cells in complete medium supplemented with 5 mM β-glycerophosphate (Sigma–Aldrich, Merck KGaA, Darmstadt, Germany) and 4 mM CaCl_2_ to achieve maximal mineralization. The calcification medium was changed every 2–3 days. After 7–10 days, cells were fixed in 4% formaldehyde in PBS for 45 min at 4 °C and then they were washed in deionized water and stained with a 2% aqueous solution of Alizarin Red S (Sigma–Aldrich, Merck KGaA, Darmstadt, Germany) for 5 min. After Alizarin red S staining, the wells were washed four times with dH_2_O while gentle shaking, and the plates were then left at an angle for 2 min to facilitate removal of excess water. Stained monolayers were visualized by phase microscopy using an inverted microscope (Nikon, Tokyo, Japan). At least 8 images per well from 5 independent assays were taken and subsequently analyzed to quantify calcium deposits by imaging analysis with Adobe Photoshop.

The recovery and semi quantification of Alizarin red S staining were performed by acetic acid extraction and neutralization with ammonium hydroxide followed by colorimetric detection at 405 nm as previously described [62]. 

### 4.8. Western Blot

Tissue lysates were separated by SDS-PAGE and transferred to 0.45 μm polyvinylidene difluoride membranes (Immobilon, Millipore. Merck KGaA, Darmstadt, Germany). Blots were incubated with antibodies directed against CD36 (NB400-144), ATF6 (NBP1-40256), and CHOP (NB600-1335) purchased from Novus Biologicals (Bio-Techne LD-R&D Systems Europe Ltd., Abingdon, UK). Equal loading of protein in each lane was verified by β-actin (A5441, Sigma–Aldrich, Merck KGaA, Darmstadt, Germany).

### 4.9. ELISA

The circulating levels of soluble CD36 in plasma from subjects were measured using commercially available ELISA kits (ABE-196-02, Nordic BioSite, Täby, Sweden) in accordance with the manufacturer’s instructions.

### 4.10. Immunostaining 

Human tissue samples of carotid plaques from patients with and without diabetes were fixed in 4% paraformaldehyde/0.1 M PBS (pH 7.4) for 24 h and embedded in paraffin. For immunohistochemistry, tissue sections (5 μm) were deparaffinized in xylene, rehydrated in graded ethanol, and treated with 0.3% hydrogen peroxide for 30 min to block peroxidase activity. Then, samples were blocked with 10% of normal serum and incubated with an antibody against CD36 overnight at 4 °C. After washing, samples were incubated for 1 h with a biotinylated secondary antibody (Vector Laboratories, Peterborough, UK). After rinsing 3 times in PBS, standard Vectastain (ABC) avidin–biotin peroxidase complex (Vector Laboratories, Peterborough, UK) was applied, and the slides were incubated for 30 min. Color was developed using 3,3′-diaminobenzidine (DAB), and sections were counterstained with hematoxylin before dehydration, clearing, and mounting. Negative controls, in which the primary antibody was omitted, were included to test for non-specific binding. Results were quantified and expressed as a percentage of positive area versus total area in independent sections of carotid plaques.

### 4.11. Determination of 7-ketocholesterol in Plasma

Plasma was separated from EDTA blood collection tubes within 2 h of the blood collection. Plasma was frozen in aliquots at –80 °C promptly after separation. Absolute quantification of 7-ketocholesterol (7-KC) was performed by liquid chromatography coupled to tandem mass spectrometry (LC-MS/MS) with atmospheric pressure chemical ionization (APCI) interface (UHPLC-(+)APCI-MS/MS) in the Centre for Omic Sciences (Reus, Tarragona, Spain), as previously described [63]. 

### 4.12. Statistical Analysis

GraphPad Prism 4.0 software (GraphPad, San Deigo, CA, USA) and R statistical software, version 3.3.1, were used for statistical analysis. Data were expressed as mean ± SEM, and values of *p ≤* 0.05 were considered significant. When data fitted a normal distribution, differences between the two groups were assessed using the Student’s *t*-test (two-tailed) and one-way ANOVA and the Bonferroni test for more than two groups. When normality failed, we used the Mann–Whitney rank-sum test to compare two groups and the Kruskal–Wallis one-way analysis of variance on ranks for multiple comparisons (Dunn’s method). The descriptive statistics of the mean (standard deviation) or median [interquartile range] of subject characteristics were estimated for quantitative variables with a normal or non-normal distribution, respectively. For the qualitative variables, absolute and relative frequencies were used. The normal distribution was analyzed by the Shapiro–Wilks test. The significance of the differences in qualitative variables was assessed by the Chi-squared test or Fisher’s exact test.

## 5. Conclusions

The scavenger receptor CD36 expression and signaling in VSMC is poorly characterized. Our findings revealed a decisive role for the CD36 scavenger in vascular calcification under hyperglycemic conditions and indicated potential mechanistic insights into the acceleration of atherosclerosis in diabetic subjects. We demonstrated that chronic exposure of VSMC to HG can accelerate inflammation and calcification in response to atherogenic stimulus through the induction of CD36 scavenger receptor. ER stress induction may be a mechanism by which CD36 signaling contributes to diabetic atherosclerosis. Altogether, our results indicate that CD36 and ER stress are potential therapeutic targets to alleviate vascular calcification in diabetes.

## Figures and Tables

**Figure 1 ijms-21-07360-f001:**
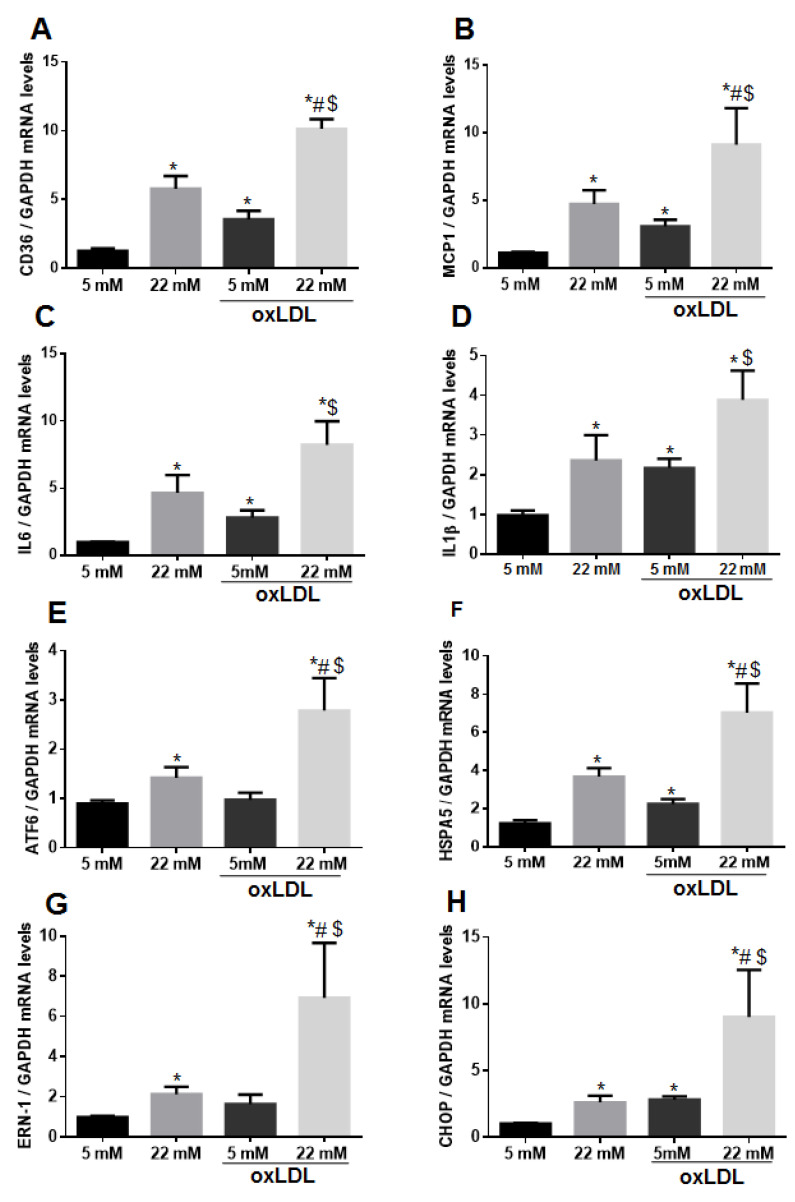
High glucose and oxidized low-density lipoprotein (oxLDL) synergize in increasing CD36 expression in vascular smooth muscle cells (VSMC). Starved VSMC were cultured for 72 h with M199 medium containing 5 mM (normal glucose, NG) or 22 mM of glucose (high glucose, HG) and treated or not with 50 μg/mL oxLDL for the last 24 h. (**A**–**H**) CD36, MCP1, IL6, IL-1β, and endoplasmic reticulum (ER) stress markers (ATF6, HSPA5, ERN-1 and CHOP) mRNA levels were determined by quantitative real-time PCR analysis (qPCR) and normalized to GAPDH. VSMC from at least four different donors were used. Mean fold change relative to untreated VSMC-NG ± SEM from five independent experiments performed in duplicate are shown. * *p* ≤ 0.05 vs. 5 mM; # *p* ≤ 0.05 vs. 22 mM; $ *p* ≤ 0.05 vs. 5 mM + oxLDL.

**Figure 2 ijms-21-07360-f002:**
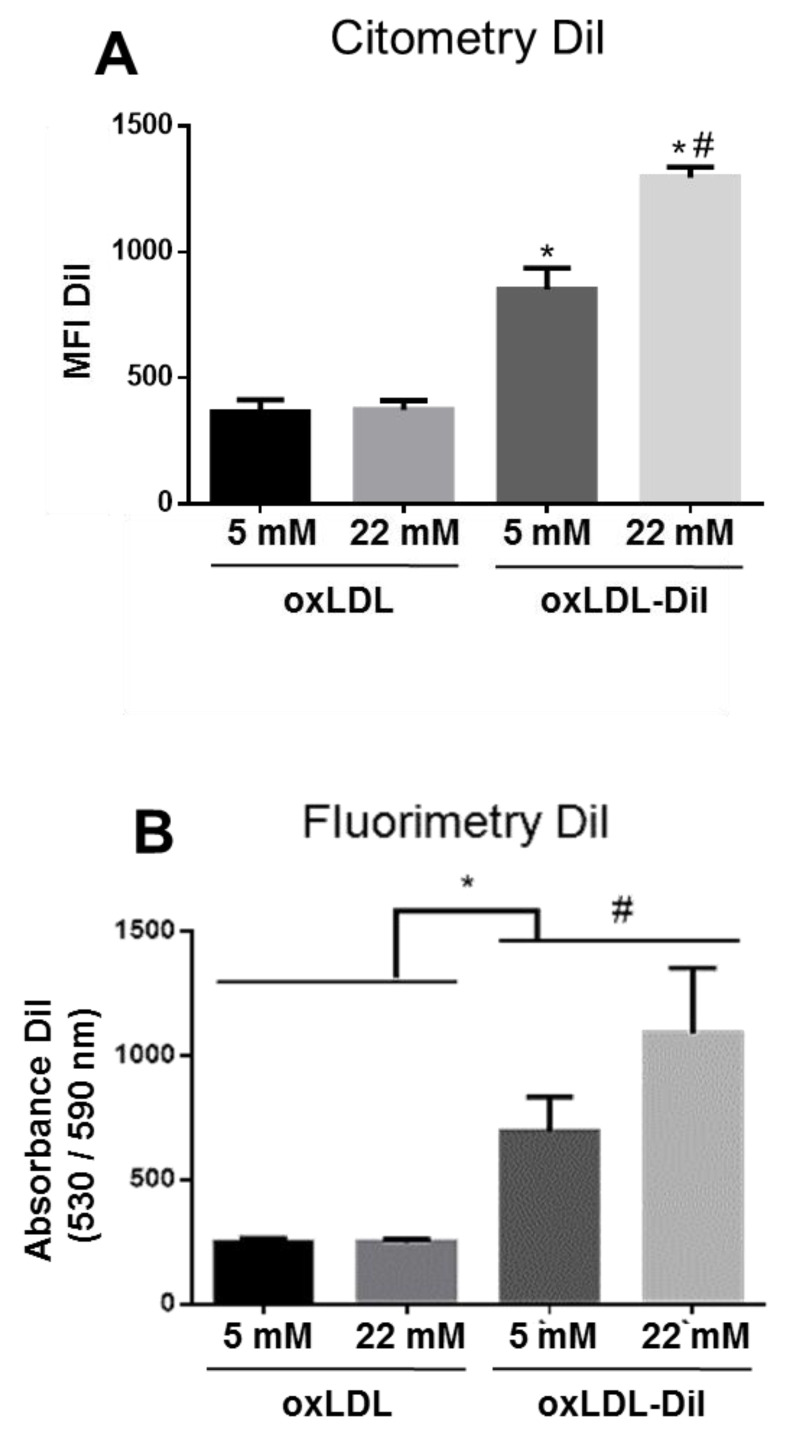
High glucose triggers oxLDL uptake by VSMC. oxLDL uptake was analyzed by flow cytometry (**A**) and by fluorimetry (**B**) in VSMC incubated for 72 h with NG or HG (22 mM) and treated with 1,1′-dioctadecyl-3,3,3′,3-tetramethylindocarbocyanine perchlorate oxLDL (DiI-oxLDL) (5 µg / mL) for the last 24 h. Cells incubated with unlabeled oxLDL were taken as control in flow cytometry and fluorimetry assays. Mean fold change relative to VSMC-NG ± SEM from four independent experiments performed in duplicate are shown. * *p* ≤ 0.05 vs. non-labeled oxLDL; # *p* ≤ 0.05 vs. 5 mM + DiI-oxLDL or 22 mM. MFI: Median fluorescence intensity.

**Figure 3 ijms-21-07360-f003:**
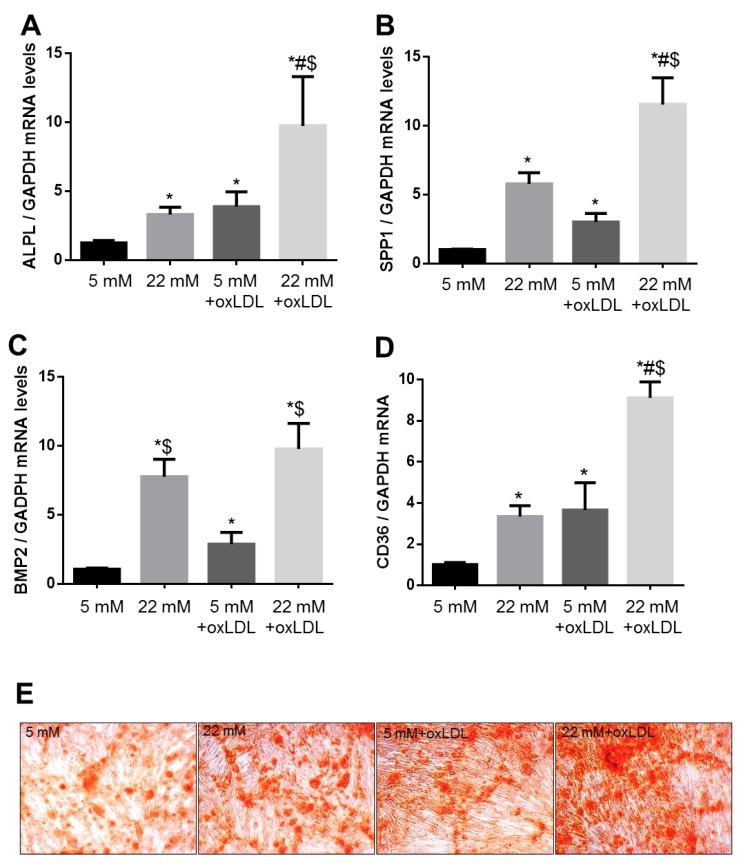
Calcification markers are induced in long-term exposed-VSMC to HG in the absence or presence of oxLDL. The expression of bone matrix proteins that regulate the calcification process was induced in VSMC cultured with high glucose for 7 days and co-incubated or not with oxLDL for the last 24 h. (**A**–**D**) Alkaline phosphatase (ALPL), secreted phosphoprotein 1 (SPP1), bone morphogenetic protein 2 (BMP2), and CD36 mRNA levels were determined by qPCR analysis and normalized to GAPDH. (**E**) Representative pictures of VSMC cultured in osteogenic medium for 7 days with NG or HG ± oxLDL and stained with alizarin red. Mean fold change relative to cultured VSMC in NG medium ± SEM (*n* = 5). * *p* ≤ 0.05 vs. 5 mM; # *p* ≤ 0.05 vs. 22 mM; $ *p* ≤ 0.05 vs. 5 mM + oxLDL.

**Figure 4 ijms-21-07360-f004:**
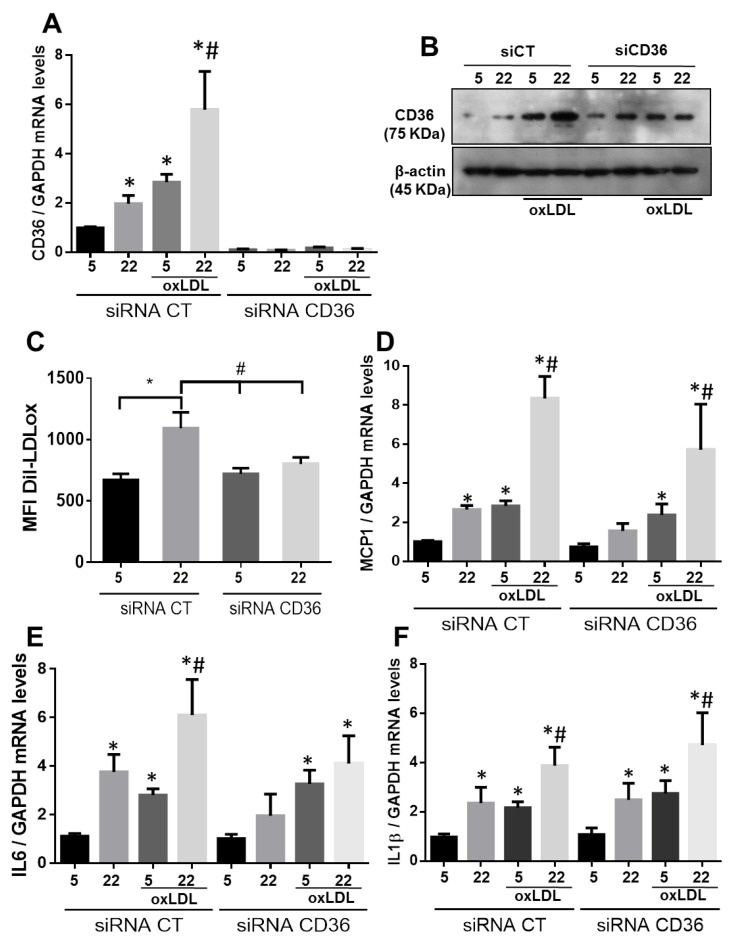
Silencing of CD36 gene expression blunts oxLDL uptake by VSMC but does not affect inflammatory markers expression. (**A**,**B**) Histogram showing the quantification of CD36 mRNA levels (*n* = 5) and representative blot of CD36 protein (*n* = 3) in VSMC transfected with siRNA-CD36 or siRNA-CT. (**C**) Quantification of DiL-oxLDL uptake by flow cytometry in VSMC transfected with siRNA-CT or siRNA-CD36 (*n* = 4). (**D**–**F**) MCP-1, IL6, and IL-1β mRNA levels were determined by qPCR analysis and normalized to GAPDH in VSMC transfected with siRNA-CT or siRNA-CD36. Mean fold change relative to VSMC transfected with siRNA-CT cultured under NG conditions ± SEM (*n* = 5). **p* ≤ 0.05 vs. 5 mM siRNA-CT or 5 mM siRNA-CD36; # *p* ≤ 0.05 vs. 22 mM siRNA-CT or 22 mM siRNA-CD36.

**Figure 5 ijms-21-07360-f005:**
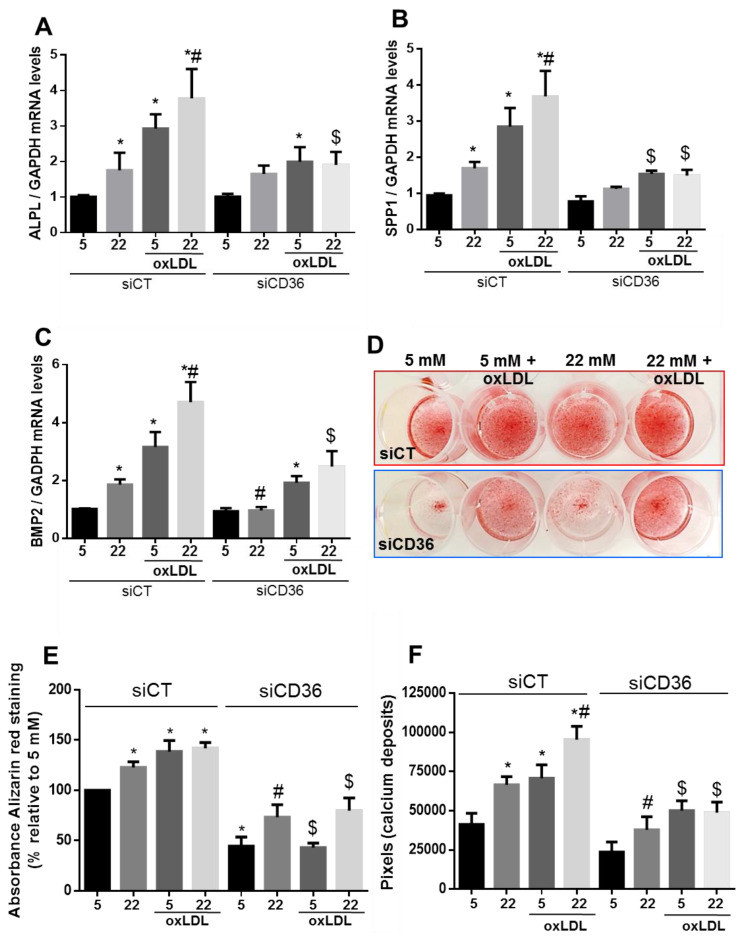
Silencing of CD36 reduces the effect of HG and oxLDL on VSMC calcification. (**A**–**C**) ALPL, BMP2, and SPP1 mRNA levels quantified by qPCR analysis in VSMC cultured with NG or HG conditions and treated or not with oxLDL after silencing CD36 (*n* = 5). (**D**) Representative picture of VSMC stained with alizarin red that were transfected with siRNA-CT or siRNA-CD36 and cultured in an osteogenic medium for 7 days with NG or HG ± oxLDL. (**E**) Quantification of alizarin red staining by colorimetry (*n* = 5). (**F**) Quantification of alizarin red staining by imaging analysis (*n* = 5). Results are expressed as mean ± SEM. *****
*p* < 0.05 vs. 5 mM siRNA-CT or siRNA-CD36; # *p* < 0.05 vs. 22 mM siRNA-CT; $ *p* < 0.05 vs. 22 mM + oxLDL siRNA-CT or 5 mM + oxLDL siRNA-CT.

**Figure 6 ijms-21-07360-f006:**
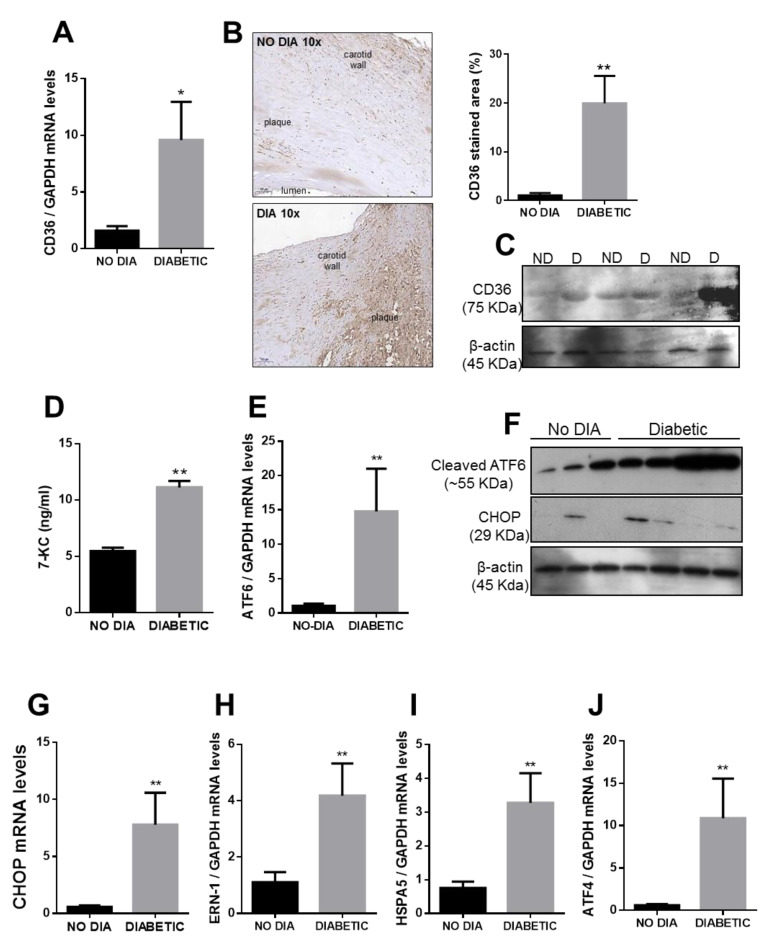
CD36 and ER stress markers expression in carotid plaques from subjects with and without diabetes. (**A**,**E**,**G**–**J**) the mRNA levels of CD36 and endoplasmic reticulum (ER) stress markers were determined in sections of carotids plaques from subjects with and without diabetes by qPCR analysis (*n* = 15). (**B**) Immunostaining of CD36 in paraffin sections of carotid plaques (*n* = 5). (**C**) Representative blot of CD36 in protein lysates from carotid plaques. (**D**) Histogram showing circulating levels of 7-KC in plasma from subjects with diabetes (*n* = 18) and subjects without diabetes (*n* = 18) determined by LC-MS/MS. (**F**) Representative blots of ATF6 and CHOP in protein lysates from carotid plaques. Results are expressed as mean ± SEM. *****
*p* < 0.05 vs. Subjects without diabetes (NO DIA). ******
*p* < 0.01 vs. NO DIA.

**Table 1 ijms-21-07360-t001:** Clinical characteristics of the study groups.

	No Diabetes *n* = 18	Diabetes *n* = 18	*p* Overall
Sex, men	16 (88.9%)	16 (88.9%)	1.000
Age, years	73.5 [68.0;79.5]	76.0 [71.0;79.8]	0.427
Hypertension	10 (55.6%)	16 (88.9%)	0.063
Dyslipidemia	16 (88.9%)	18 (100%)	0.486
BMI, kg/m^2^	25.8 (3.47)	27.4 (3.76)	0.188
Smoking:			0.582
No	3 (17.6%)	5 (27.8%)	
Yes	4 (23.5%)	2 (11.1%)	
Former smoker	10 (58.8%)	11 (61.1%)	
sBP, mmHg	140 [130;148]	130 [113;145]	0.145
dBP, mmHg	78.7 (8.31)	70.5 (11.7)	0.022
Antiplatelet treatment	14 (77.8%)	17 (94.4%)	0.338
Statin treatment	16 (88.9%)	16 (94.1%)	1.000
Glucose, mg/dL	94.5 [85.0;111]	133 [119;160]	0.001
Creatinine, mg/dL	0.94 (0.39)	0.98 (0.37)	0.784
eGFR, mL/min/173 m^2^	82.8 [60.5;90.0]	79.0 [57.0;90.0]	0.647
ALT, U/L	18.0 [16.0;30.8]	20.5 [15.5;26.8]	0.727
GGT, U/L	35.0 [17.0;52.0]	23.0 [13.0;42.0]	0.208
Triglycerides, mg/dL	86.0 [72.8;105]	102 [81.0;142]	0.227
Total cholesterol, mg/dL	138 [123;147]	112 [102;125]	0.028
HDL cholesterol, mg/dL	43.0 (11.7)	39.7 (9.23)	0.422
LDL cholesterol, mg/dL	80.0 [57.0;93.0]	59.0 [45.0;73.0]	0.099
CRP, mg/L	13.1 [2.20;21.3]	4.76 [1.15;8.75]	0.118
Glycated hemoglobin, %	5.70 [5.50;5.95]	6.70 [6.12;7.50]	<0.001
Glycated hemoglobin, mmol/mol	39.0 [37.0;41.5]	50.0 [43.2;58.0]	<0.001
C peptide, nmol/L	1.02 [0.90;1.64]	1.17 [0.90;1.46]	0.760
Leukocyte count, ×10^9^/L	8.85 [6.68;9.53]	8.20 [6.27;9.85]	0.743
Hemoglobin, g/dL	13.0 (1.81)	12.2 (1.32)	0.152
Hematocrit, %	39.4 (5.51)	36.5 (3.76)	0.081
Platelets, ×10^9^/L	195 (52.2)	192 (41.5)	0.882
Albuminuria, mg/L	20.4 [5.45;49.9]	4.40 [2.70;24.4]	0.174
Albumin/creatinine ratio, mg/g	13.9 [5.20;50.5]	11.4 [3.80;24.7]	0.765
Diabetes duration, years	-	12.2 (3.43)	-

Data are shown as median [interquartile], means (SD) or n (%). BMI, body mass index; sBP, systolic blood pressure; dBP, diastolic blood pressure; eGFR; estimated glomerular filtration rate; ALT, alanine aminotransferase; GGT, Gamma-glutamyl transferase; CRP, C-reactive protein.

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
