# Peer review of "Role of the Scavenger Receptor CD36 in Accelerated Diabetic Atherosclerosis"

_ijms, 2020, doi:10.3390/ijms21197360_

Round 1
Reviewer 1 Report
This is a potentially interesting contribution which explores the effect of elevated glucose concentration on the expression of CD36 by cultured human vascular smooth muscle cells, derived from the donors from multi-organ donors. The focus on aortic SMC is appropriate as many of the cells of the atheroma are of smooth muscle cell origin. They have also examined the pro-inflammatory gene expression induced by elevations in medium glucose. All of these effects are enhanced by exposure to 5ug of OxLDL. The role of CD36 is further examined by silencing the expression of the receptor. Finally osteogenic genes were examined to try and explain the increasing calcification observed in diabetic arteries. To increase the relevance of these observations to the in vivo human pathology, they have also examined carotid endarterectomy samples from patients with or without diabetes.
Despite the potential interest of these findings, there are a number of issues that remain to be addressed, particularly the mechanistic relationship between glucose, CD36 and OxLDL. It is clear that not all observations can be explained by the effect of CD36 on the uptake of OxLDL.
1The mechanisms can be more thoughtfully explored. The authors have examined the response of smooth muscle cells to varying concentrations of glucose. In examinig the enhancing influence of OxLDL why did they not also examine the effects of varying concentrations of the OxLDL? A careful examiniaton of their data shows that the relationships are complex. Does one need to use 5ug of yhe lipoprotein to observe its effect or is there a threshold level that elicits the response. As stated not all effecs are artributable to increased uptake of OxLDL. Assuming that CD36 is the main mediator of OxLDL uptake the results on some genes when CD36 is silenced are unchanged.
2. With point#1 in mind, what is the relationship between glucose signaling and OxLDL signaling. CD36 is one fulcrum. BUt what else might be at play?
3. What is the basis of CD36 induction by high concentrations of glucose?
4. Figure 2 employs DiI -OxLDL with the DiI being used as a marker. But is this all it is? If the exoeriment were done with DiI native LDL,what would we see? Or if the experiment were done with a mixture of DiIOxLDL (0.5ug) mixed with 4.5ug of OxLDL, would the same result be seen?
5. The silencing experiments depicted in figure 4 are problematic, The silencing appears to be very efective for the CD36 transcript but not for the protein. The protein is reduced to control levels when exposed to OXlDL, but nothig lije the transcript reduction. This raises the questionn of two pools of CD36 of differing stability. In the light of these complexities, how does one explain the relatively minor changes in MCP-1 expression compared to the unsilenced situation
6. Given the results shwn infigures 4 and 5 together does oe conclude that the induction of osteogenic genes occurs independently of the induction of pro-inflammatory genes? Discuss.
In summary, despite the interest of the observations they reveal complexities that are not captured in the discussion
Author Response
We thank the reviewers for the thorough review and suggestions, which we believe have allowed us to strengthen the manuscript. We have edited the text and provided detailed responses to each point raised below (indicated by * before each response and red font). Please, see the attached file answers to reviewers IJMS Navas-Madroñal et al. pdf for an extended version of the responses including additional figures.
Reviewer #1:
This is a potentially interesting contribution which explores the effect of elevated glucose concentration on the expression of CD36 by cultured human vascular smooth muscle cells, derived from the donors from multi-organ donors. The focus on aortic SMC is appropriate as many of the cells of the atheroma are of smooth muscle cell origin. They have also examined the pro-inflammatory gene expression induced by elevations in medium glucose. All of these effects are enhanced by exposure to 5ug of OxLDL. The role of CD36 is further examined by silencing the expression of the receptor. Finally osteogenic genes were examined to try and explain the increasing calcification observed in diabetic arteries. To increase the relevance of these observations to the in vivo human pathology, they have also examined carotid endarterectomy samples from patients with or without diabetes.
Despite the potential interest of these findings, there are a number of issues that remain to be addressed, particularly the mechanistic relationship between glucose, CD36 and OxLDL. It is clear that not all observations can be explained by the effect of CD36 on the uptake of OxLDL.
- The mechanisms can be more thoughtfully explored. The authors have examined the response of smooth muscle cells to varying concentrations of glucose. In examinig the enhancing influence of OxLDL why did they not also examine the effects of varying concentrations of the OxLDL? A careful examiniaton of their data shows that the relationships are complex. Does one need to use 5ug of yhe lipoprotein to observe its effect or is there a threshold level that elicits the response. As stated not all effecs are artributable to increased uptake of OxLDL. Assuming that CD36 is the main mediator of OxLDL uptake the results on some genes when CD36 is silenced are unchanged.
*Answer 1: we thank the reviewer for his / her suggestion of varying concentrations of the oxLDL; however, we used the concentration of 50 ug / mL oxLDL because it is widely described in literature in VSMC and macrophages culture assays (1-7). Moreover, other authors previously used higher concentrations of oxLDL such as 100 ug / mL to perform experimental procedures with VSMC and reported that increasing the concentration of oxLDL over 50 ug / mL did not elicit further response in dose-response assays (8-11). We did not vary the concentrations of oxLDL in the cell culture but extended the time of exposure to oxLDL till 48 h and determined by MTT assay that cell viability was decreased in human primary VSMCs. This is explained because Ox-LDL induce apoptosis in smooth muscle cells (12). We chose 24 h of incubation with purified oxLDL based on literature and on the increased toxicity caused by longer exposure in our primary cell culture. Please, see below the histogram showing the results of MTT cell viability assay (N=5).
- Cell Calcium. 2020 12;91:102265; 2. J Cell Biochem. 2016 Nov;117(11):2496-505; 3. FASEB J. 2014 Nov;28(11):4779-91; 4. Leukoc Biol. 2014 Mar;95(3):509-20; 5. Toxicol Appl Pharmacol. 2013 Dec 15;273(3):651-8; 6. J Lipid Res. 2003 Sep;44(9):1667-75; 7. Biochem Biophys Res Commun. 2013; 437(1):62-6. 8. Mol Cell Biochem. 2012 Mar;362(1-2):115-22; 9. Cell Death Dis. 2014 Apr 17;5(4):e1182; 10. Int J Biochem Cell Biol. 2015 May;62:54-61; 11. Mol Med Rep. 2015 Jun;11(6):4341-4;12. Biochem Biophys Res Commun. 1996 Jun 14;223(2):413-8.
.* In regards to the effects associated to increased uptake of OxLDL elicited by HG, we agree with the reviewer, not all of them can be atributable to the oxLDL uptake. This is a functional assay aiming to prove that HG elicit oxLDL uptake by VSMC through CD36 receptor. The effects observed by the exposure to HG with or without the presence of oxLDL cannot be explained by the unique action of CD36 since we cannot assume that CD36 is the only mediator of oxLDL uptake. Indeed, in VSMC, there are other receptors for lipids that may be up-regulated in VSMC by oxLDL that can uptake oxLDL such as lectin-like oxidized LDL receptor-1 (LOX-1), LDL receptor, low density lipoprotein receptor-related protein (LRP1) and class A scavenger receptor (SR-A) (1-5). However, under long term exposure to HG we have observed that silencing of CD36 abolishes oxLDL uptake indicating the importance of this particular receptor in human VSMCs. In addition and supporting our results, previous studies demonstrate that advanced glycation end-products (AGEs), a direct effect of chronic exposure to HG, may play a pathogenic role in the vascular calcification process (6).
- J Biol Chem. 2000; 275(23):17661-70; 2. Arterioscler Thromb Vasc Biol. 2002; 22(3):387-93; 3. Inflammation 2014; 37(2):555-65; 4. J Biol Chem. 2015; 290(24):14852-65; 5. Antioxidants (Basel). 2019; 8(7):218; 6. Atherosclerosis. 2012; 221(2):387-96.
We discussed this issue in the revised manuscript and added new references 36-40. Please, see discussion section page 11, lines 244-251.
- With point#1 in mind, what is the relationship between glucose signaling and OxLDL signaling. CD36 is one fulcrum. BUt what else might be at play?
*Answer 2. We thank the reviewer for this interesting question. In the present work we are intending to mimic the chronic exposure of vascular cells to high glucose and hyperlipidemia that occur in diabetic subjects suffering from atherosclerosis in order to elucidate why atherosclerosis is accelerated in diabetes. The major cell types involved in atherogenesis, macrophages and VSMC, are activated by pro-inflammatory stimuli including modified LDL. In literature, it is well documented that high plasma low density lipoprotein (LDL) levels become atherogenic when oxidized to modified LDL (Ox-LDL) by inducing foam cell formation via enhanced CD36 expression on macrophages which is prevalent in vascular lesions. In addition to Ox-LDL, raised levels of glucose, insulin resistance, low HDL cholesterol and increased levels of free fatty acid (FFA); all result in increased expression of CD36, thereby contributing to T2DM and related atherosclerosis (1). Specifically, in type 2 diabetes, the interplay between cells and inflammatory mediators up-regulates CD36 expression in macrophages whereas the advanced glycation end products (AGE) induces CD36 expression in aortic vascular smooth muscle cells (VSMCs). Together with increased plasma AGEs, aortic RAGE overexpression and increased oxidative stress markers induce CD36 overexpression in diabetes (2).
- Mol Genet Metab. 2011; 102(4):389-98;2. Nutr Metab Cardiovasc Dis. 2008;18(1):23-30.
According to the reviewer’s comment, we discussed this issue in the revised manuscript and added new references 42 and 43. Please, see discussion section, page 11, lines 254-259.
- What is the basis of CD36 induction by high concentrations of glucose?
*We thank the reviewer for this interesting question. Proatherogenic factors relevant to human diabetes, including high glucose, oxLDL, advance glycation end products, and C-reactive protein enhance receptors for ox-LDL, primarily in the endothelial cells, that allows uptake of ox-LDL into vascular cells (1,2). Moreover, CD36 expression has been previously reported to be altered by hyperglycaemia in atherosclerotic patients (3). In the present study, we have determined the effect of HG on CD36 expression in human aortic VSMC and the synergisitic effect with oxLDL on inflammation, vascular calcification and oxLDL uptake. We have edited the discussion part and included additional information regarding factors related to diabetes that are involved in CD36 induction and references 44 and 45. Please, see page 11, lines 254-261.
- Cardiovasc Res. 2006; 69(1):36-45;2. Circ Res. 2004; 95(9):877-83; 3. Eur J Clin Invest. 2011; 41(8):854-62.
- Figure 2 employs DiI -OxLDL with the DiI being used as a marker. But is this all it is? If the exoeriment were done with DiI native LDL,what would we see? Or if the experiment were done with a mixture of DiIOxLDL (0.5ug) mixed with 4.5ug of OxLDL, would the same result be seen?
*Answer 4. Oxidized low density lipoprotein (ox-LDL), a marker of oxidative stress, is present in the plasma as well as in the atherosclerotic arteries of patients with atherosclerosis. Ox-LDL leads to endothelial activation, dysfunction and injury. Scavenger receptor-mediated uptake of oxidized LDL (oxLDL) is thought to be the major mechanism of foam cell generation in atherosclerotic lesions. However, native LDL is also capable of contributing to foam cell formation (1-3). In literature, most of the studies where the uptake of nativeLDL is described, are performed in macrophages but only few studies reported so far the uptake of native-LDL uptake labelled with DiI by VSMC (4-6). Since we did not run additional assays using native LDL, we do not know if the uptake of DiI-nLDL would be similar in VSMC or not. It is an important question that will be explored in further studies.
- Arterioscler Thromb Vasc Biol. 2010; 30(10): 2022-31; 2. J Lipid Res. 2012; 53(10):2081-91; 3. J Lipid Res. 2014 Aug;55(8):1648-56; 4. Circulation. 2004; 110(4):452-9; 5. J Biol Chem 2015; 290(24):14852-65; 6. Mater Chem B. 2019; 7(16):2703-2713.
*In response to the second question regarding figure 2, and as described in methodology, we performed a competition analysis by using an excess of unlabeled oxLDL that was added into the medium 1 h before the addition of 5 μg/ml DiI‐oxLDL. Cellular uptake of Dil oxLDL was confirmed by competition with an excess amount (50 μg/ mL) of unlabeled human oxLDL. We did not include the results of the competition test in the histograms of figure 2. Please, see below the histogram with the results obtained by FACS analysis and including the test competition analysis.
- The silencing experiments depicted in figure 4 are problematic, The silencing appears to be very efective for the CD36 transcript but not for the protein. The protein is reduced to control levels when exposed to OXlDL, but nothig lije the transcript reduction. This raises the question of two pools of CD36 of differing stability. In the light of these complexities, how does one explain the relatively minor changes in MCP-1 expression compared to the unsilenced situation
*Answer 5. We understand reviewer’s concern about CD36 protein expression in silencing experiments. It is known that the CD36 scavenger receptor turnover is low, like it happens with many receptors, and thus, it is very difficult to abolish protein expression completely with a transient gene silencing. Also, we formulate the hypothesis that the long exposure to HG may trigger the stabilization of the mRNA encoding CD36 allowing a sustained expression of the protein. In support to our hypothesis, there is evidence in literature about the implication of post-transcriptional regulatory mechanisms that increase or decrease stabilization of mRNA encoding LDL receptors. This would explain the differential expression of the gene compared with the protein (1,2). Further molecular analysis need to be achieved to confirm this hypothesis. Finally, we would like to remark that CD36 silencing is significantly reducing the effect of oxLDL on CD36 protein expression.
Regarding the second question that the reviewer arises, we think that silencing of CD36 is not affecting to the expression of pro-inflammatory markers because the exposure to HG and oxLDL may modulate the inflammatory pathways in different ways in VSMCs and not only through the CD36 receptor signaling. As described in literature, HG triggers oxidative stress generation, induces cellular senescence and leads to ER stress induction and cell apoptosis; altogether, these effects are related to the activation of pro-inflammatory pathways in vascular cells (3-7). However, we would like to remark the effect of silencing CD36 on the expression of calcification markers triggered by HG alone or in combination with oxLDL (Figure 5) as a potential mechanism that can explain why the process of vascular calcification in atherosclerosis is exacerbated in diabetic subjects.
- Genes (Basel). 2020; Jul 29;11(8):E861; 2. J Biol Chem. 2003; 278(10):7884-90. 3. Atheroscler Thromb. 2001; 8(2):55-62; 4. Diabetologia. 2005; 48(7):1401-10; 5. Eur J Nutr. 2007; 46(8):431-8; 6. PLoS One. 2014; 9(2):e88391; 7. Clin Sci (Lond). 2018;132(6):719-738.
We added new references and discussed the lack of effect of CD36 silencing on inflammatory markers expression. Please, see discussion section, page 11, lines 268-276.
*Additionally, we studied whether the increased expression of ATP binding cassette transporter A1 (ABCA1), which mediates the cellular efflux of phospholipids and cholesterol to lipid-poor apolipoprotein A1 (apoA1), by HG and oxLDL was affected by the CD36 silencing. We observed that silencing of CD36 mildly reduced the induced-ABCA1 mRNA levels by HG when combined with oxLDL. Animal and in vitro experiments suggest that ABCA1 not only mediates cholesterol and phospholipid efflux, but is also involved in the regulation of apoptosis and inflammation (Soumian et al. Vasc Med. 2005;10:109-119). Our results evidence that HG and oxLDL do not only affect CD36 scavenger receptor but also to other proteins implicated in intracellular lipids homeostasis in VSMC, independently of CD36 expression. Please, see below the histogram showing the unpublished result of ABCA1 expression
- Given the results shown infigures 4 and 5 together does not conclude that the induction of osteogenic genes occurs independently of the induction of pro-inflammatory genes?. Discuss.
*Answer 6. We agree with the reviewer at this point. In this work we aimed to determine whether CD36 receptor signalling contributes to HG and Ox-LDL-induced differentiation and calcification of human VSMCs in vitro. Vascular calcification is a major feature of advanced atherosclerosis and highly associated with cardiovascular diseases. In literature, it is well documented that ox-LDL) is a critical risk factor for atherosclerosis and osteogenic differentiation of vascular smooth muscle cells (VSMCs). Ox-LDL independently induce osteogenic differentiation of vascular smooth muscle cells (VSMCs) through the increase in oxidative stress and alkaline phosphatase (ALP) activity (1,2). Inflammatory cytokines are produced mainly by monocyte/macrophages and lymphocytes, but also by endothelial cells and smooth muscle cells (SMC) after stimulation by inflammatory mediators. Several cytokines such as TNFα, IL-6 and IL-1β exert potent pro-inflammatory effects in atherosclerosis and other metabolic and inflammatory disorders (3-6). In atherosclerotic lesions, oxLDL participates from the induction of the gene expression of cytokines and adhesion molecules in VSMCs (7-10). However, there are no studies describing the combined effects of HG and oxLDL on VSMC mineralization. Furthermore, in our study downregulation of CD36 expression using small interfering RNA reduced Ox-LDL-induced VSMC mineralization either in cells cultured with low glucose (5 mM) or with high glucose (22 mM).
- Eur J Pharmacol. 2017;794:45-51; 2. Arterioscler Thromb Vasc Biol. 2011 Mar;31(3):608-15; 3. Atherosclerosis, 2005;180:11-17; 4. Circulation, 1998;97:242-244; 5. Cardiovasc. Res. 2008;79:360-376; 6. Arterioscler. Thromb. Vasc. Biol, 1999;19:2364-2367; 7. J Biol Chem 2007; 282:19167-19176; 8. Thromb Haemost 2008; 100:1155-1165; 9. Am J Physiol Heart Circ Physiol 2009; 296: H987-H996; 10. Molecular and Cellular Biochemistry 2012; 362: 115–122.
We discussed this issue in the discussion section. Please, see page 12, lines 290-294.
- In summary, despite the interest of the observations they reveal complexities that are not captured in the discussion.
As suggested by the reviewer we improved and extended the discussion in some parts. Please, see pages 11 and 12 in discussion section and new added references in pages 19 and 20. All the changes are in red font.

Reviewer 2 Report
The authors found that oxLDL-induced hyperglycemia and vascular calcification in diabetes is dependent on CD36 expression. The authors of the study noted that vascular smooth muscle cells in the presence of high glucose concentration and oxLDL produce significant amounts of cytokines, calcification markers, ER stress markers, well as well increase CD36 expression. Moreover, the results were confirmed in carotid plaques from diabetic patients compared to nondiabetic patients.
I do not have major comments. The investigations were performed according to high quality standard of research. I was delighted to read this manuscript which includes multidirectional research. I congratulate the authors of their excellent research.
Nevertheless, I have some open questions:
- I know you can buy a VSMC cell line. Why did the authors choose to isolate cells for their research?
- What about the control cells? Did you determine parameters in cells without the addition of glucose and oxLDL? Do "hungry" cells induce the production of inflammatory markers and markers of oxidative stress?
Author Response
We thank the reviewer for the thorough review and suggestions. We have provided detailed responses to each point raised below (indicated by * before each response and red font).
Reviewer 2.
The authors found that oxLDL-induced hyperglycemia and vascular calcification in diabetes is dependent on CD36 expression. The authors of the study noted that vascular smooth muscle cells in the presence of high glucose concentration and oxLDL produce significant amounts of cytokines, calcification markers, ER stress markers, well as well increase CD36 expression. Moreover, the results were confirmed in carotid plaques from diabetic patients compared to nondiabetic patients.
I do not have major comments. The investigations were performed according to high quality standard of research. I was delighted to read this manuscript which includes multidirectional research. I congratulate the authors of their excellent research.
Nevertheless, I have some open questions:
1.I know you can buy a VSMC cell line. Why did the authors choose to isolate cells for their research?
*Answer 1: in response to reviewer’s concern about human cells source, we know that commercial primary human VSMC are available in ATCC repository (Primary Aortic Smooth Muscle Cells; Normal, Human (HASMC) (ATCC® PCS-100-012™). However, these cells are extracted from the aortic tissue of one subject and because of this, inter-individual variability is lost. Thus, we think it is a more realistic approach to use primary cultures from different donors (5-6) in our assays and in order to mimick the complexity of every subject biology.
- What about the control cells? Did you determine parameters in cells without the addition of glucose and oxLDL? Do "hungry" cells induce the production of inflammatory markers and markers of oxidative stress?
*Answer 2: we took as control those cells cultured in basic M199 medium which contains 5 mM of glucose. Please, see that in the composition of M199 medium (Ref. 11150059, GIBCO, Thermo Scientific) where low glucose is specified and it is equivalent to 5 mM according to the manufacturer.
In response to the second question, serum starvation before you start a cell based assay is a common procedure. The reason it is performed is to synchronize all your cells to the same cell cycle phase and make cells behave more accordingly. However, we did not completely starved the cells since we performed all the experiments with VSMC in the presence of a reduced amount of serum (1 %). Further experiments need to be achieved to determine the production of inflammatory markers and oxidative stress in VSMC under these specific stimuli and in the presence of higher amount of serum.
Round 2
Reviewer 1 Report
The manuscript is modestly improved. The authors have attempted to respond to my earlier critique, but in no case have they done further experiments. Their modifications have involved further literature citations and sentences added to the discussion. But the manuscript does not adequately reflect the complexity of the role of CD36. Nor does the discussion though reasonable, deal with the limitations of their research.
1.They have studied smooth muscle cells exclusively, but other cells in the plaque may contribute. So the role of CD36 as studied here can only reflect on its role in this cell type.
2. The basis of the choice of 50ug of OxLDL appears to be based on the literature. What is the in vivo concentration of OxLDL? and how does this relate to these phenomena? They report on the co-operation of high glucose and OxLDL in promoting CD36 expression. The previous question was whether in the presence of high glucose the cells may be particularly sensitive to OxLDL, i.e. to a lower concentration of OxLDL than that used for the reported experiments.
3.In response to my previous point #5 the authors conclude that despite the elimination of CD36 message by silencing the retention of substantial levels of CD36 protein is most likely due to the stability of the protein. This may be but is it purely co-incidental that the protein level does decline somewhat but just to the control level? The stability issue could perhaps be addressed by extending the duration of the experiment. Even if the stability of the protein is the issue, it still seems possible that there are two pools of CD36 of differing stability. This may help to explain the next issue.
4. The response of inflammatory proteins(eg MCP-1) and osteogenic proteins differs in response to CD36 silencing, the former being more resistant than the latter. This suggests that the pathways regulating the expression of these genes are different. The authors have made relatively few comments on pathways, even based on the literature
5. The last sentence of the abstract overstates the role of CD36. It may be partially involved in the phenomena reported in this manuscript.
Author Response
We thank the reviewer for the thorough review and suggestions. We apologize for not having addressed properly some of the reviewer’s questions in the first round. We have edited the text and provided detailed responses to each point raised below (indicated by * before each response and the manuscript has been edited including the changes in red font).
The manuscript is modestly improved. The authors have attempted to respond to my earlier critique, but in no case have they done further experiments. Their modifications have involved further literature citations and sentences added to the discussion. But the manuscript does not adequately reflect the complexity of the role of CD36. Nor does the discussion though reasonable, deal with the limitations of their research.
- They have studied smooth muscle cells exclusively, but other cells in the plaque may contribute. So the role of CD36 as studied here can only reflect on its role in this cell type.
* We agree with the reviewer. We are aware that other cells types such as macrophages and endothelial cells contribute to the formation and progression of the atherosclerotic plaque. The role of CD36 scavenger receptor in macrophages and endothelial cells on plaque formation under high glucose exposure or under atherogenic conditions is already described in the literature (Gautam et al. Mol Genet Metab. 2011 Apr;102(4):389-98; Wang Z, et al. Atherosclerosis. 2012; 221(2):387-96; Hayashi T, et al. PLoS One. 2014;9(2):e88391; Méndez-Barbero et al., EMBO Mol Med. 2013;5(12):1901-1917); however, to our knowledge, few studies focused on the role of CD36 in VSMC in diabetes or atherosclerosis (de Oliveira Silva C,et al. Nutr Metab Cardiovasc Dis. 2008;18(1):23-30). As mentioned in the introduction section, VSMC-derived foam cells in human atherosclerotic plaque comprise as many as 50% of foam cells in human and murine lesions. The uptake of oxLDL by VSMCs leads to foam cell formation, apoptosis and secretion of chemokines that contribute to the recruitment of inflammatory cells. Extracellular lipid pools colocalize with αSMA+ cells already in early phases of the atherosclerosis; however, the implication of CD36 in oxLDL internalization and inflammatory response by VSMC has not been elucidated under hyperglycemic conditions. Thus, we focused our work on the role of CD36 specifically in VSMC biology under hyperglycemic conditions alone or in combination with oxLDL.
We discussed this issue in the revised manuscript and added the new references 29-32. Please, see discussion section: page 11, lines 233-239.
- The basis of the choice of 50ug of OxLDL appears to be based on the literature. What is the in vivo concentration of OxLDL? and how does this relate to these phenomena? They report on the co-operation of high glucose and OxLDL in promoting CD36 expression. The previous question was whether in the presence of high glucose the cells may be particularly sensitive to OxLDL, i.e. to a lower concentration of OxLDL than that used for the reported experiments.
* The plasmatic concentration of LDL in humans is around 1 mg / mL. The percentage of oxLDL of total LDL is estimated at the range between 1 - 5 %. Moreover, the plasmatic oxLDL levels vary according to the technique used for their quantification. However, the circulating oxLDL levels are not related to local oxLDL concentrations found in plaque tissue. These local levels of oxLDL determine the paracrine and autocrine effects of oxLDL on different cells located in the plaque and in those cells surrounding it. So far, the concentration of oxLDL in the atherosclerotic plaque or in the arterial wall has not still been determined. We chose the concentration of 50 µg / mL oxLDL based on the previous literature because it is the most widely used concentration described not only in VSMC culture assays but also in macrophages culture. We agree with the reviewer that other authors previously used smaller concentrations of oxLDL such as 12.50 - 40 ug / mL to perform experimental procedures with VSMC; nevertheless, in some cases, the exposure to oxLDL was extended to 48 h (Yang T, et al. Cell Calcium. 2020; 91:102265; Chien MW, et al. J Lipid Res. 2003. doi: 10.1194/jlr.M300006-JLR200). With our results, we cannot certainly state that exposure to high glucose may predispose VSMC to be particularly sensitive to oxLDL and, actually, it may be possible that a lower concentration of oxLDL may have worked. The question raised by the reviewer opens an interesting pathway of research: if exposure to HG may predispose VSMC to be particularly sensitive to oxLDL, it would favour the development of plaque and accelerate the atherosclerotic process. This can partially explain why atherosclerosis is accelerated in diabetes.
- In response to my previous point #5 the authors conclude that despite the elimination of CD36 message by silencing the retention of substantial levels of CD36 protein is most likely due to the stability of the protein. This may be but is it purely co-incidental that the protein level does decline somewhat but just to the control level? The stability issue could perhaps be addressed by extending the duration of the experiment. Even if the stability of the protein is the issue, it still seems possible that there are two pools of CD36 of differing stability. This may help to explain the next issue.
* As the reviewer points out, retention of substantial levels of CD36 protein after silencing is most likely due to the stability of the protein and as we stated before, because of the long lifespan of the CD36 scavenger receptor. Honestly, we cannot assess if it is possible the co-existance of two pools of CD36 with different stability but it could explain the discrepancy between the protein expression and the mRNA levels. Extending the duration of the experiment is not advisable since the CD36 silencing is performed by a transient transfection of the siRNA pool and the effect may be lost after 72 h. We would like to remark that even when the protein expression is only partially blocked compared to non-silenced controls, we still observe how CD36 silencing affect oxLDL uptake and pro-calcification markers expression.
- The response of inflammatory proteins (eg MCP-1) and osteogenic proteins differs in response to CD36 silencing, the former being more resistant than the latter. This suggests that the pathways regulating the expression of these genes are different. The authors have made relatively few comments on pathways, even based on the literature
* We agree with the reviewer. We apologize for not having addressed the reviewer’s question properly. We are hereby doing our best to clarify this point and improve this part of the manuscript by discussing this differential effect of CD36 silencing.
The receptor for advanced glycation end-products (RAGE) has been suggested to play a pivotal role in the development of diabetic vasculopathy and atherosclerosis. Recent studies demonstrate that the induction of MCP-1 expression may occur through the activation of receptor of advance glycation end-products (RAGE) signaling via NF-κB and mitogen-activated protein kinase (Dwarakanath et al. J Mol Cell Cardiol. 2004;36(4):585-95; Hayakawa et al. J Atheroscler Thromb. 2012;19(1):13-22). In the same line, other authors described how human aortic VSMCs exposed to high glucose (HG) (diabetic milieu) or S100B, a ligand of the receptor for advanced glycation end products, exhibited significantly increased binding of THP-1 monocytic cells through MCP-1 (Meng et al. Am J Physiol Heart Circ Physiol. 2010;298:H736-745).
On the other hand, oxLDL promotes a chronic inflammatory process by activating the MAPKinases signaling pathways (MAPK, PI3 K/Akt, TAT/JAK) in macrophages and in vascular cells, which in turn trigger the activation of transcription factors (NFkB, AP1, PPAR) involved in endothelial dysfunction, cytokines and chemokynes secretion and apoptosis (Ross R. N Engl J Med. 1999;340:115-26; Navab et al. J Lipid Res. 2004;45:993-1007). The above-described works support our data and help us to explain why CD36 silencing does not modulate pro-inflammatory markers expression and indicate that the effects of exposure to HG on MCP-1 expression may occur through the activation of RAGE and not through CD36 receptor derived signaling.
We discussed this issue in the revised manuscript and added new references 40-47 and 55-56. Please, see discussion section: pages 11-12, lines 263-275 and lines 289-293.
- The last sentence of the abstract overstates the role of CD36. It may be partially involved in the phenomena reported in this manuscript.
* In response to the reviewer suggestion, the last sentences of the abstract were modified to down-tune our conclusions. Please, see abstract section, lines 37-39.

Round 3
Reviewer 1 Report
In reference to my previous review, there are three points that require further attention.
- The authors response to my previous point #2 there isva need to address this experimentally by using lower concentrations of OxLDL in the presence of high and low glucose. It is important to know whether high glucose sensitizes smooth muscle cells to OxLDL.
- The question of two pools of CD36 is not satisfactorily addressed. Do the authors know that the silencing would not be effective at 72 hours of incubation? Why is it that the residual CD36 is not functional in the uptake of OxLDL and pro-calcification marker expression. Does this suggest that the residual CD36 is not fully functional? Can the authors test another function of CD36--eg fatty acid uptake to assess functionality.
- The study is apparently driven by an interest in atherosclerosis. But in the context of atherosclerosis in vivo it is not clear that the conclusions of these studies is relevant in view of the widespread expression of CD36 in other cells and in view of the possible participation of other receptors in responding to high glucose and OxLDL. The potential participation of CD36 in atherosclerosis is still not fully stablished and can probably be only furthered by selective cell specific knockout animals for atherosclerosis studies. These complexities and limitations should be reflected in the conclusion.
Author Response
In reference to my previous review, there are three points that require further attention.
- The authors response to my previous point #2 there isva need to address this experimentally by using lower concentrations of OxLDL in the presence of high and low glucose. It is important to know whether high glucose sensitizes smooth muscle cells to OxLDL.
Answer: in the present work, we focused on the effects of chronic exposure to HG on CD36 expression and activity in VSMC and, additionally, studied also the synergistic effects of oxLDL in this setting. Overall, and keeping in mind the current situation of pandemia with the difficulties associated, it does not make sense to us performing additional experiments with varying the concentrations of oxLDL.
- The question of two pools of CD36 is not satisfactorily addressed. Do the authors know that the silencing would not be effective at 72 hours of incubation? Why is it that the residual CD36 is not functional in the uptake of OxLDL and pro-calcification marker expression. Does this suggest that the residual CD36 is not fully functional? Can the authors test another function of CD36--eg fatty acid uptake to assess functionality.
Answer: we did not raise the question of two pools of CD36; actually, it was the reviewer who proposed this explanation. As previously answered in the second round of questions, we observed that CD36 silencing blocked the induction of CD36 gene expression. This resulted in the reduction of CD36 protein expression which in turn was reflected in the expression of the calcification markers and oxLDL uptake.The objective of this work was not to analyze the turn-over / kinetics of CD36 protein degradation. This matter goes beyond the scope of the manuscript.
- The study is apparently driven by an interest in atherosclerosis. But in the context of atherosclerosis in vivo it is not clear that the conclusions of these studies is relevant in view of the widespread expression of CD36 in other cells and in view of the possible participation of other receptors in responding to high glucose and OxLDL. The potential participation of CD36 in atherosclerosis is still not fully stablished and can probably be only furthered by selective cell specific knockout animals for atherosclerosis studies. These complexities and limitations should be reflected in the conclusion.
Answer: this time, we do not completely agree with the reviewer’s point of view. Our study was focused on the potential role of CD36 in diabetes and its effects on vascular calcification and diabetic accelerated atherosclerosis. We do not see the point of performing in vivo experiments with ApoE-/- or LDLr-/- animals (as atherosclerosis animal models) to study the potential participation of CD36, since it has been already described and addressed in the literature (Absence of CD36 protects against atherosclerosis in ApoE knock-out mice with no additional protection provided by absence of scavenger receptor A I/II. Kuchibhotla S, et al. Cardiovasc Res. 2008;78(1):185-96; The role of the scavenger receptor CD36 in regulating mononuclear phagocyte trafficking to atherosclerotic lesions and vascular inflammation. Harb D, et al. Cardiovasc Res. 2009;83(1):42-51. doi: 10.1093/cvr/cvp081; A major role for RCAN1 in atherosclerosis progression. Méndez-Barbero N, et al. EMBO Mol Med. 2013;5(12):1901-17. doi: 10.1002/emmm.201302842;). Moreover, in the present work there is an entire figure including the results obtained from human samples, serum and carotid plaques from diabetic and non-diabetic subjects, where we showed the increased expression of CD36 in tissue from diabetic subjects compared to non-diabetic subjects. These results suggest the importance of CD36 in diabetic accelerated atherosclerosis and vascular calcification.